# Transcriptional Profiling Reveals Ribosome Biogenesis, Microtubule Dynamics and Expression of Specific lncRNAs to be Part of a Common Response to Cell-Penetrating Peptides

**DOI:** 10.3390/biom10111567

**Published:** 2020-11-17

**Authors:** Tomas Venit, Moataz Dowaidar, Maxime Gestin, Syed Raza Mahmood, Ülo Langel, Piergiorgio Percipalle

**Affiliations:** 1Science Division, Biology Program, New York University Abu Dhabi (NYUAD), Abu Dhabi P.O. Box 129188, UAE; tv21@nyu.edu (T.V.); srm632@nyu.edu (S.R.M.); 2Department of Biochemistry and Biophysics, Stockholm University, Svante Arrhenius väg 16B, SE-106 91 Stockholm, Sweden; moataz.dowaidar@gmail.com (M.D.); maxime.gestin@dbb.su.se (M.G.); ulo.langel@dbb.su.se (Ü.L.); 3Laboratory of Molecular Biotechnology, Institute of Technology, University of Tartu, Nooruse 1, 50411 Tartu, Estonia; 4Department of Molecular Biosciences, The Wenner-Gren Institute, Stockholm University, SE-106 91 Stockholm, Sweden

**Keywords:** CPPs, cell-penetrating peptides, ribosome, rRNA ribosome biogenesis, microtubule, centrosome, long noncoding RNA, lncRNA, penetratin, PepFect14, mtCPP1, TP10

## Abstract

Cell-penetrating peptides (CPPs) are short peptides that are able to efficiently penetrate cellular lipid bilayers. Although CPPs have been used as carriers in conjugation with certain cargos to target specific genes and pathways, how rationally designed CPPs per se affect global gene expression has not been investigated. Therefore, following time course treatments with 4 CPPs-penetratin, PepFect14, mtCPP1 and TP10, HeLa cells were transcriptionally profiled by RNA sequencing. Results from these analyses showed a time-dependent response to different CPPs, with specific sets of genes related to ribosome biogenesis, microtubule dynamics and long-noncoding RNAs being differentially expressed compared to untreated controls. By using an image-based high content phenotypic profiling platform we confirmed that differential gene expression in CPP-treated HeLa cells strongly correlates with changes in cellular phenotypes such as increased nucleolar size and dispersed microtubules, compatible with altered ribosome biogenesis and cell growth. Altogether these results suggest that cells respond to different cell penetrating peptides by alteration of specific sets of genes, which are possibly part of the common response to such stimulus.

## 1. Introduction

Cell-penetrating peptides (CPPs) are usually short oligopeptides consisting of 5–30 amino acids. Despite the high variability in their chemical structure and conformation they share the ability to penetrate cell membrane without the involvement of energy-dependent processes [1]. Transfer through membranes together with their ability to bind and deliver various cargos into cells made them interesting tools for drug delivery. The first cell penetrating peptide was derived from the trans-activator of transcription (HIV-1 TAT) protein of the human immunodeficiency virus in 1988 [2], followed by many others in subsequent years. Further to these initial observations, a lot of efforts were placed to understand how they are internalized in cells, how they are attached to cargos and, possibly, how to achieve cell type specificity and subcellular localization needed for clinical use [3,4]. For CPP internalization, two main mechanisms have been proposed-direct translocation through the cellular membrane and energy dependent endocytosis, each providing several models of CPP uptake mostly dependent on their chemical structure and conformation (for review see [5]). Cargos can be attached to CPPs covalently through direct interaction or via transporting systems such as polymer carriers, liposomes, metal nanoparticles or triazol-based originating from a “click” reaction [6,7,8,9,10]. Non-covalent binding uses electrostatic interactions between a positively charged peptide and negatively charged cargo and can be used for transportation of siRNA into the cells [11]. Cell type specificity, subcellular localization and clinical use of CPPs are interconnected as CPP bound drugs need to be delivered to specific subset of cells and to specific cellular compartment. For example, in anti-cancer drug delivery, nucleus and mitochondria are the most targeted with the DNA damaging drugs or nucleic acid-based gene therapy approaches [12]. The lack of cell type specificity of CPPs remains a major obstacle in clinical development but recent development of cancer targeting peptides which use the unique characteristics of the tumor microenvironment could improve cancer cell specificity [13]. An example is provided by the acidity-triggered rational membrane (ATRAM) peptide which uses the mild acidity of tumor tissue [14] to penetrate the membranes [15].

The ability of organisms to survive depends on their capacity to respond to change and to adapt to new conditions. Organisms have developed several systems that enable dynamic adaptation to immediate stresses and changes within the environment. Rearrangement of chromatin and subsequent changes in gene transcription are among the first to respond to external stimuli, followed by changes in translation and functional changes at the cellular level such as cytoskeletal rearrangement or organelle organization [16]. Even though CPPs have been shown to be generally well tolerated and not toxic to cells in small concentration [1], they are actively and passively transported through the membranes to the cellular interior. Although this is likely to elicit cellular responses, studies have been mostly focusing on cell survival and toxicity and it is not known how and whether specific sets of genes are targeted by CPPs within a temporal frame.

To establish how global transcription responds to CPP treatment, in this study we performed a comprehensive profiling of cells treated with different sets of CPPs. Using RNA-seq on RNA samples isolated from CPP-treated HeLa cells through time course experiments, we identified different sets of differentially expressed genes. Results from these analyses indicate that genes involved in ribosome biogenesis, centrosome and microtubule dynamics are the most affected. Using an image-based high content phenotypic profiling platform and cell proliferation assay, we found that differences in gene expression correlate with phenotypic alterations which are likely to be compatible with misregulated growth, proliferation and cell cycle progression. Interestingly, expression of specific long-noncoding RNAs is affected after CPP treatment, suggesting that they are also part of a specific common response pathway with a significant impact on cellular metabolism.

## 2. Material and Methods

### 2.1. Preparation of Cell Penetrating Peptides

Penetratin, mtCPP1, PF14 and TP10 were synthesized using a Biotage Alstra+ microwave assisted synthesizer (Biotage, Uppsala, Sweden) on a 0.1 mmol scale following a fluorenylmethyloxycarbonyl (Fmoc) solid phase peptide synthesis chemistry protocol. The first amino acid of each peptide was attached to a ring-amide Chemmatrix resin (0.5 mmol/g) in order to obtain a C-terminal amide group. The resin was swollen at 70 °C in dimethylformamide (DMF) for 5 min with the oscillating mixer on. The amino acids were dissolved in DMF containing 2-cyano-2-(hydroxyimino)acetate (Oxyma pure) and N,N′-diisopropylcarbodiimide (DIC) and added to the resin for 5 min at 75 °C. The Fmoc protective groups were removed by the action of 20% piperidin in DMF for 2 min at 45 °C followed by 12 min at room temperature. For PF14 synthesis, the stearic acid was added under the same condition as the amino acids. The final cleavage was performed by trifluoroacetic acid (TFA) (95%), water (2.5%) and triisopropylsilane (TIS) (2.5%) for 4 h at room temperature. The peptides were extracted from the cleavage cocktail by precipitation in cold diethyl ether. The crude peptides were purified by reverse phase high performance liquid chromatography using a Biobasic C8 column with a gradient of acetonitril and water supplemented with 0.1% TFA. The mass of each peptide was verified by UHPLC-MS (Agilent 1260 Infinity, Agilent Technologies, Santa Clara, CA, USA). The peptides were then freeze dried. Penetratin, mtCPP1, PF14 and TP10 stock solutions were prepared by dissolving each peptide in 1xPBS to a final concentration of 40 µM. For the treatment, each peptide stock solution was mixed with full DMEM media to a final 2 µM peptide concentration.

### 2.2. Tissue Culture and Peptide Treatment

HeLa cells (ATCC^®^ CCL-2) were cultured in DMEM with high glucose, 10% fetal bovine serum, 100 U/mL penicillin and 100 mg/mL streptomycin (all from Millipore-Sigma, Burlington, MA, USA), in a humidified incubator with 5% CO_2_ at 37 °C. For peptide treatment, 5 × 10^5^ cells were seeded in 6-well plates with 2 mL of full-DMEM medium 24 h before the treatment. Next day, the medium was exchanged with pre-warmed DMEM medium supplemented with respective peptides and incubated for defined times. As a negative control, HeLa cells were incubated with pre-warmed DMEM medium supplemented with 1× PBS without any of the CPPs.

Antibodies against fibrillarin (ab4566), coilin (ab87913), Sc35 (ab11826), tubulin (ab195883), 58K (ab27043) and M6PR (ab124767) were all purchased from Abcam (Cambridge, MA, USA). Alexa Fluor 555 Goat Anti-Rabbit (ab150078) and Alexa Fluor 488 Goat Anti-Mouse (ab150117), used as secondary antibodies, were also purchased from Abcam. Hoechst 43222 (H1399) were purchased from Invitrogen (Waltham, MA, USA).

### 2.3. High Content Phenotypic Profiling

Cells were cultured in 96-well clear bottom assay plate (Corning, Corning, NY, USA) at a density of 20,000 cells/well and treated with penetratin for 1, 4, or 8 h respectively. Following treatment, cells were fixed with 4% formaldehyde for 10 min, and permeabilized with 0.5% Triton X-100 for 10 min. Cells were then incubated with primary antibodies overnight and washed three times with phosphate-buffered saline with 0.5% Tween-20 (PBST buffer). Next, cells were incubated with fluorochrome-conjugated secondary antibodies for 1 h and with Hoechst (5000× dilution) for 20 min, washed twice with PBST and stored in PBS buffer. The plate was analyzed via the Cellomics ArrayScan XTI High-Content Screening (HCS) platform (Thermo Fisher Scientific, Waltham, MA, USA). The image analysis was performed using the Compartment Analysis Bio Application software (Thermo Fisher Scientific). Hoechst stained nuclei were used for defining primary objects. Fluorescence signal for each staining was quantified in the form of spots, which can be defined as any discrete punctate objects that fall within measuring area and intensity thresholds. Depending on the staining specificity of the antibody and type of the stained cellular compartment, these spots does not need to fully overlap with the stained organelle and represent quantification of any fluorescent signal found in the dedicated area. As it is image-based analysis, the size and intensity measurements were done in pixels and final graphs show the ratios between different conditions in arbitrary units. Fluorescent measurements were quantified per well. Eight wells per condition were used in each experiment, each containing at least 5000 cells. Each experiment was repeated three times with similar results.

### 2.4. RNA-Seq Library Preparation, Sequencing, and Analysis

Total RNA from Hela cells untreated or treated with each of the above mentioned CPPs during the time course was extracted using the TRI Reagent (Millipore-Sigma) from two biological replicates per condition. The RNA-seq library was prepared using the TruSeq RNA Library Prep Kit v2 (Illumina, San Diego, CA, USA) and sequenced with the HiSeq 2500 next generation sequencing platform (performed at the NYUAD Sequencing Center). All of the subsequent analysis, including quality trimming, was executed using the BioSAILs workflow execution system. The raw reads were quality trimmed using Trimmomatic (version 0.36) to trim low quality bases, systematic base calling errors, as well sequencing adapter contamination. FastQC was used to assess the quality of the sequenced reads pre/post quality trimming. Only the reads that passed quality trimming were retained for downstream analysis. The quality trimmed RNAseq reads were aligned to the human GRCh38 genome using HISAT2 (version 2.0.4). The resulting SAM alignment files for each sequenced sample were then converted to BAM format and sorted by coordinate using SAMtools (version 0.1.19). The BAM alignment files were processed using HTseq-count using the reference annotation file to produce raw counts for each sample. To reduce the possible noise in the data caused by differential expression of low count genes, we discarded all DE genes from subsequent analysis which have joint count of raw reads in both replicates less than 50 (corresponding to a minimum CPM value of 1), representing 3.6% of DE genes for 1 h Penetratin treatment, 4.01% for 4 h Penetratin treatment, 7.27% for 4 h mtCPP1 treatment, 6.6% for 4 h TP10 treatment and 7.6% for 4 h PF14 treatment. The raw counts were then analyzed using the online analysis portal NASQAR in order to merge, normalize and identify differentially expressed genes. Differentially expressed (DE) genes (log2(FC) ≥ 0.5 and adjusted *p*-value of <0.05 (Benjamini-Hochberg) for upregulated genes and log2(FC) ≤ −0.5 and adjusted *p*-value of <0.05 for downregulated genes) between the treated and untreated Hela cells were subjected to Gene Ontology (GO) analysis with standard background using DAVID Bioinformatics and gene enrichment groups were plotted based on their significance (Benjamini-Hochberg adjusted *p*-value). Clustering and visualization of temporal expression patterns was performed using the R package TCseq, version 1.8.0. Raw counts for negative control samples and samples treated with penetratin for 1 h or 4 h were filtered to retain genes with CPM values of 1 or more in at least two conditions. Genes showing log2 fold change of more than 0.5 relative to negative control at each timepoint with adjusted *p*-value of less than 0.05 (Benjamini-Hochberg) were selected for clustering. Z-score transformed counts of selected genes were hierarchically clustered into 4 groups using correlation as a distance metric with the command timeclust(tca, algo = “hc”, k = 4, standardize = TRUE, dist = ‘correlation’). Cluster plots were generated using the command timeclustplot. To generate heatmap of z-scores for Penetratin-based clusters across all CPPs, raw counts for all treatments were filtered to retain genes with CPM >1 in at least 2 conditions followed by log2 transformation and Z-score scaling. Heatmap of Z-scores for genes present in different clusters was plotted using the package ComplexHeatmap [17]. Venn diagram was produced by Bioinformatics & Evolutionary Genomics platform. RNA-seq data were deposited in the Gene Expression Omnibus (GEO) database under accession number GSE148689.

### 2.5. Quantitative RT-PCR

Total RNA was extracted using RNazol (Sigma, St. Louis, MO, USA) according to the manufacturer’s instructions. RNA (500 ng) per sample was cleaned from residual gDNA by Turbo DNA-free kit (Ambion, Austin, TX, USA) and reverse transcribed using RevertAID First Strand cDNA Synthesis Kit (Thermo Fisher Scientific). Diluted cDNA was subjected to quantitative real-time PCR analysis using Maxima SYBR Green qPCR Mix (Thermo Fisher Scientific) and the three-step cycling protocol on the StepOne Plus Real-Time Thermal Cycler (Applied Biosystems, Foster city, CA, USA). Relevant primers were designed against the selected genes (Appendix A). qPCR analysis was performed in triplicates and in two different experiments with similar results. The expression data were analyzed by normalizing each sample to the GAPDH expression.

### 2.6. Cell Proliferation Assay

Hela cells were seeded in 96-well plates to 20% confluence (5000 cells/well) and grown overnight. Cells were treated with penetratin or 1× PBS as described above and media containing penetratin or 1× PBS was exchanged every 24 h before measurement. The cell proliferation was measured by CellTiter-Blue^®^ Cell Viability assay (Promega, Madison, WI, USA) according to the manufacturer’s protocol. On a given time, 20 µL of CTB reagent was added to each well and incubated at 37 °C for 2 h, to allow cells to reduce resazurin to fluorescent resorufin. Cell proliferation was quantified by measuring total fluorescence at 560 ex/590 em nm using Synergy H1 microplate reader (BioTek, Winooski, VT, USA). For time point “0 h”, penetratin or 1× PBS containing media was added to cells just before adding CTB reagent and incubated for 2 h before measurements. Four replicates were used for each condition and each time point.

### 2.7. Statistical Analysis

GraphPad Prism 8.3.0 software (GraphPad, San Diego, CA, USA) was used for statistical analysis. Unpaired *t*-test was used in all the experiments, unless otherwise stated in text. Error bars in boxplots represents minimum and maximum values. Error bars in bar charts represents SD. Red dots in bar charts represents each measured value. Statistical significance is marked with * *p* < 0.05, ** *p* < 0.01, *** *p* < 0.001, **** *p* < 0.0001, ns (not significant). Each experiment was performed at least in triplicates. Exact description of sample sizes and number of replicates is defined for each experimental procedure described in the Methods section.

## 3. Results

### 3.1. Transcriptional Profiling Reveals Time Dependent Cellular Response to CPPs

To investigate time-dependent changes in transcriptional patterns in response to CPP treatment, we performed RNA-seq on Hela cells treated with the CPP penetratin for 1 or 4 h, respectively, and compared to untreated cells. PCA analysis and distance heatmap of RNA-seq data with the distribution of each sample replicates shows significant differences in the expression profiles of the three groups (Figure 1A). Interestingly, samples treated for 4 h with penetratin show global gene expression patterns to be more similar to untreated cells than to cells subjected to 1-h treatment. This suggests that cells tend to respond immediately to CPPs by changing their transcriptional profile but this acute response changes over time. Pairwise comparisons of gene expression of (1) untreated (NC) and treated cells for 1 h (penetratin 1 h), (2) untreated (NC) and treated cells for 4 h (penetratin 4 h), or (3) cells treated for 1 h (penetratin 1 h) and treated for 4 h (penetratin 4 h) respectively, show that there is significant amount of differentially expressed genes between each group (Figure 1B, Appendix A). To find out how the transcriptional landscape changes in time in response to treatment with penetratin, we performed a differential analysis between treated and untreated cells and hierarchically clustered the genes that were differentially expressed relative to control cells after both 1 and 4 h of penetratin exposure (Figure 1C, Appendix A). This analysis identified four unique gene clusters that showed distinct temporal gene expression patterns upon penetratin treatment. Genes in two of these clusters (1 and 4) showed a strong transcriptional response after 1 h of penetratin exposure but partially reverted to their basal expression levels after 4 h. These results suggest strong initial response of cells to penetratin which fades over time as these two clusters contain the majority of differentially expressed genes. However, the genes in the second set of clusters (2 and 3) showed a relatively modest initial response with more pronounced transcriptional changes appearing only after 4 h of penetratin treatment (Figure 1C) suggesting some specific pathways and gene groups to be affected later in the treatment.

Seeing that there are time dependent changes in gene expression after penetratin treatment, we next subdivided differentially expressed genes into early response genes (up- or down-regulated only in the samples treated with penetratin for 1 h), late response genes (up- or down-regulated only in the samples treated with penetratin for 4 h) and common genes (up- or down-regulated in samples treated with penetratin for 1 and 4 h). Specific gene groups which show opposite expression profile between 1- and 4-h treatment are marked as genes with switched expression (Figure 1D and Appendix A). To validate the observed RNA-seq data, we selected specific genes with highly changed expression belonging to common response group, which react to penetratin by either decreasing or increasing their expression upon both, 1- and 4-h treatments and genes whose expression switches between the timepoints (Appendix A) and performed quantitative RT-PCR (Appendix A).

We included additional timepoints for 8- and 12-h to see if the observed expression profile persists over time. Quantitative RT-PCR data correlate with the RNA-seq data, especially for the group of genes whose expression is heavily increased upon 1 h treatment but normal or decreased upon 4 h (Appendix A). The prolonged incubation of cells with penetratin has variable effects on gene expression of selected genes as some preserve or even strengthen their expression level through the whole treatment, while in others, the expression profile is restored to original levels (Appendix A). Next, we performed gene ontology (GO) analysis on all differentially expressed genes between untreated samples and samples treated for 1 or 4 h (log2(FC) ≥ 0.5 and adjusted *p*-value of <0.05 for upregulated genes and log2(FC) ≤ −0.5 and adjusted *p*-value of <0.05 for downregulated genes) and analyzed them using DAVID Bioinformatics software (Appendix A). We found two main groups of genes overrepresented in the analysis, those related to ribosome biogenesis and translation, and those related to microtubule dynamics, centrosome structure and cell cycle. Gene ontology analysis of only up- or only down-regulated genes (Figure 2 and Figure 3, Appendix A) revealed an overrepresentation of GO terms related to rRNA processing, ribosome biogenesis and translation only in the group of up-regulated genes and overrepresentation of microtubule related GO terms only in the down-regulated genes. While there are some specific GO terms related to early (1 h) or late (4 h) response only, ribosome biogenesis and microtubule dynamics seem to be affected through the whole treatment.

To test whether the observed transcriptional profiles are specific only to penetratin treatment or are part of a more general response of cells to CPPs, we treated cells with an additional set of CPPs-mtCPP1 [18], transportan 10 (TP10) [19] and PepFect14 (PF14) [20]. MtCPP1 is a short 4 amino acid long CPP that cannot bind and transport cargoes. However, this peptide is able to target mitochondria and, thanks to its dimethyltyrosine residues, it works as scavenger of reactive oxygen species, reducing their intracellular levels [18].

PF14 is derived from TP10 [20], but has unique properties and, in complex with oligonucleotides, it aggregates to form nanoparticles with a hydrodynamic diameter ranging in 100 nm [21]. We incubated HeLa cells for 4 h with each of the above peptides (including penetratin) and transcriptionally profiled cells by RNA-seq. The PCA analysis as well as distance heatmap of all samples reveal significant differences between untreated samples, samples treated with penetratin for 1 h or samples treated with each of the 4 CPPs for 4 h (Figure 4A,B, Appendix A). All samples treated for 4 h with each CPP cluster together (Figure 4A), although PCA analysis suggests some differences among CPPs, PF14 being the most different (Figure 4C). These differences are further evidenced by Venn diagrams plotting all differentially expressed genes after 4 h of CPP treatment and showing overlay between each cell penetrating peptide (Figure 4D). Compatible with the fact that these 4 CPPs have distinct structures, properties, mechanisms of uptake, intracellular localization and different physicochemical properties, each treatment leads to a peptide-specific transcriptional response. However, the majority of differentially expressed genes is in the intersection of all 4 peptides, suggesting that cells exhibit a common primary transcriptional response to all CPPs regardless of their different structures and properties. We then asked if the early and late response gene clusters identified on the basis of penetratin treatment also responded to treatment with other CPPs in a similar manner and analyzed the expression of genes in each cluster after 4 hours’ exposure to PF14, TP10 and mtCPP1. Strikingly, our results showed that the three different CPPs induced similar gene expression changes in each cluster as those observed after penetratin treatment. It is therefore likely that these changes represent a conserved cellular response to CPPs (Figure 4E). This is further supported by the GO analysis of the genes found in the intersection of 4 CPPs (Appendix A) showing similar GO terms related to Ribosome biogenesis among upregulated genes and microtubule-related genes among downregulated genes (Appendix A).

### 3.2. Increased Nucleolar Size Suggests Increased Levels of Ribosome Biogenesis Upon Penetratin Treatment

Next, we merged all up- and down-regulated genes in samples treated with penetratin for 1 and 4 h which are related to ribosome and ribosome biogenesis and plotted them as a heatmap of normalized counts (Figure 5A,C). Expression of genes related to ribosome biogenesis is gradually increasing over time (Figure 5A) while expression of ribosomal proteins is most prominent after 1 h treatment with penetratin (Figure 5C).

Differentially expressed genes cover all steps through ribosome biogenesis, starting with regulation of RNA Polymerase I transcription (TCOF1, CIRH1A, HEATR1, WDR43), cleavage of ribosomal RNA (IMP3, IMP4, NAT10, UTP18, RCL1, MPHOSPH10), catalytic activity during ribosome biogenesis (NHP2, EMG1, GAR1, WDR36, NOP58), nuclear export (NXT5, EIF6, RAN, RIOK1, RIOK2) and biogenesis of mature tRNA (RPP25, POP7, POP1, RPP40) (Figure 5B). Genes encoding structural proteins affected by penetratin treatment are overrepresented in large and small ribosomal subunits in both, nuclear and mitochondrial ribosomes (Figure 5D). The rate of ribosome production is regulated mainly at the level of rRNA synthesis occurring in the nucleolus and intense ribosome biogenesis is usually reflected in the increased size of nucleoli [22]. As we revealed a general upregulation of ribosomal and non-ribosomal proteins present in the nucleolus, we quantified number and size of nucleoli by image-based high-content phenotypic profiling using an anti-fibrillarin antibody (Figure 6A). Staining was quantified with the Compartmental Analysis BioApplication software inbuilt in the high content screening platform and at least 20000 cells were used for each measurement. In agreement with RNA seq data, we observed significant increase in fibrillarin-positive spot size, number and intensity through the treatment. Even though these spot parameters represent any fibrillarin-positive structures, so they cannot be fully corelated with nucleolar parameters, the observed phenotype supports the idea of increased rRNA synthesis in cells upon penetratin treatment given that the majority of fibrillarin normally localizes in the nucleolus (Figure 6B). These changes appear to be specific for nucleoli since the size of the nucleus itself and size and number of other nuclear compartment such as Cajal bodies and nuclear speckles, immunostained with anti-Coilin and anti-SC35 antibodies respectively, did not change upon penetratin treatment (Appendix A).

### 3.3. Misregulation of Microtubule- and Centrosome-Related Genes Upon Penetratin Treatment Is Accompanied by a Dispersion of Microtubules in Cells

Centrosomes are composed of two centrioles and large amount of proteins termed pericentriolar material (PCM). In general, the centrosome is responsible for nucleation of most of the microtubules, as well as regulation of cell-cycle progression [23]. We have shown above that microtubule-related genes are overrepresented in the downregulated group of genes after penetratin treatment. To analyze this further in detail we merged all differentially expressed genes related to microtubules, centriole and centrosome obtained in samples treated with penetratin for 1 and 4 h (Figure 7A). These genes were plotted as a heatmaps of normalized counts (Figure 7B) or as an interaction map based on experimental data and databases search (Figure 7C). We found that the vast majority of genes related to centrioles and centrosomes are downregulated after penetratin treatment. The protein products of these genes can be roughly divided into two clusters-cluster 1 containing core protein components of centrosomes and cluster 2 containing proteins which have different regulatory functions related to centrosome structure and function (Figure 7C). Microtubule related genes can be divided into two clusters as well. Cluster 3 contains structural components of microtubules including α-, β-, and γ-tubulins while cluster 4 consists of microtubule dependent motor proteins, which can be further divided into clusters 4A-kinesin and kinesin-like proteins, 4B-dynein proteins and 4C-intraflagellar transport proteins (Figure 7C). Interestingly, while microtubule-associated genes are mostly downregulated, structural α-β- and γ-tubulins are upregulated after penetratin treatment (Figure 7B). Other affected genes, not falling within any of 4 clusters have distinct regulatory or cell signaling functions related to microtubules assembly and dynamics (Figure 7C).

Next, we performed high-content phenotypic profiling of β-tubulin stained cells (Figure 8A). Similarly to the analyses reported above, measured spot parameters do not fully overlap with microtubules but only represent β-tubulin-positive spots which passed through the threshold. In untreated cells, microtubules are mostly localized in the perinuclear region around centrosome and their abundance gradually decreases towards the cell periphery. Therefore, in terms of high content phenotypic profiling, β-tubulin-positive spots are clustered together around the nucleus and show higher intensity and average size but lower numbers (as a lot of smaller, low intensity spots would cluster to make a few bigger, brighter spots). Upon penetratin treatment these parameters gradually change over time and β-tubulin-positive spot average intensity and size are reduced, while the number of spots is gradually increased, suggesting that microtubules in penetratin-treated cells lose their perinuclear localization and are more dispersed around the cells (Figure 8B). This agrees with a previous RNA-seq data set showing suppression of centrosomal genes and upregulation of some specific tubulin genes. To see whether the observed tubulin pattern is microtubule-specific or represents general changes in the cytoplasm over time, we used the same parameters to quantify p58-stained Golgi apparatus and M6PR-stained lysosomes. Both organelles show some differences upon penetratin treatment but their pattern is different from microtubules, supporting the specificity of the measurements (Appendix A).

### 3.4. Penetratin Treatment Decreases Proliferative Potential of Cells

The localization, amount and dynamics of the microtubules are important in many processes such as cell cycle and cell division, proliferation and differentiation. Centrosome serves as the main microtubule organizing center (MTOC) in proliferative and undifferentiated cells and loss of centrosomal MTOC activity is associated with differentiation and formation of non-centrosomal cell-type specific arrays of microtubules [23]. Therefore, downregulation of centrosomal genes, upregulation of some tubulin genes and reorganization of microtubules in penetratin treated samples could be explained by a shift of cells towards a quiescent phase accompanied by a decreased level of cell proliferation. To test this hypothesis, we performed CTB proliferation assay (Figure 8C) over a period of 4 days. The results show no significant change between treated and untreated cells at the beginning of the assay (0 h) proving no acute toxicity of penetratin on Hela cells. However already after 24 h, there is a significant reduction in the proliferation of penetratin treated cells which is further reinforced after 48 h. Over the time, the differences between conditions flatten as untreated cells reach confluency and autoinhibit themselves. In conclusion, penetratin treatment does not entirely block proliferation but rather it leads to a decrease in the proliferative potential of cells.

### 3.5. Treatment with Penetratin Affects Expression of Long Non-Coding RNAs

When performing gene ontology analysis of differentially expressed genes, we found many unidentified genes which were excluded from the analysis. Three quarters of the human genome is actively transcribed but only a minor part is represented by protein-coding genes [24]. Other transcripts represent enormous variety of non-coding RNAs which are generally less described and underrepresented in gene ontology analyses. Among non-coding RNAs, long non-coding RNAs (LncRNAs) are the most abundant, having a function in important biological processes such as transcriptional regulation, X chromosome inactivation, genomic imprinting and cell differentiation. LncRNAs have been also found to be dysregulated in different human diseases including cancer making them interesting target for drug development [25] Therefore, we compared all differentially expressed genes in cells treated with penetratin for 1 and 4 h with the list of annotated LncRNAs collected in LNCpedia database, version 5.2 (Figure 9 and Appendix A). Surprisingly, despite similar amounts of up-(n = 2934) and down-regulated (n = 3182) genes after penetratin treatment, there is high abundance of LncRNAs among the downregulated genes (n = 531) in comparison to upregulated genes (n = 52) (Figure 9A,B). Many of these differentially expressed LncRNAs have unknown function, however we found several LncRNAs which have been previously described (Figure 9C). The most important LncRNA found to be upregulated after penetratin treatment is H19, which is considered to be one of the major genes in cancer, actively involved in all stages of tumorigenesis [26]. Among downregulated genes, there are several small nucleolar RNA host genes as well as many cancer related LncRNAs such as MINCR-MYC-induced long non-coding RNA associated with lung cancer [27], ZEB1 antisense RNA 1 associated with osteosarcoma [28] or PURPL-p53 upregulated regulator of p53 levels which suppresses p53 and promotes tumorigenicity in colorectal cancer [29]. LncRNAs NEAT1 and NEAT2 also known as MALAT1 are among the best described non-coding RNAs in human diseases and were proposed as possible targets for anti-cancer therapies [30,31]. Other group of downregulated LncRNAs regulates nuclear architecture, gene expression and genomic stability. LncRNA HOTAIR-HOX transcript antisense RNA-regulates global gene expression by interacting with polycomb repressive complex 2 (PRC2) and histone demethylase LSD1 and is important in cancer invasiveness and metastasis progression [32]. LncRNA FIRRE modulates nuclear architecture in complex with hnRNPU [33]. Noncoding RNA NORAD-noncoding RNA activated by DNA damage-sequesters Pumilio proteins which repress the stability and translation of mRNAs for mitotic, DNA repair and DNA replication proteins [34].

## 4. Discussion

In this paper, we have used RNA sequencing and image-based high throughput analysis of CPP-treated HeLa cells to unveil the cellular pathways affected by CPP uptake. We have discovered that there is a time dependent response of cells to cell penetrating peptides, affecting same sets of genes regardless on the type of CPP. Despite some differences, RNA sequencing results show high similarity in gene expression between the different CPP treatments suggesting that there is a general response of cells to such stimuli. Whether such response is specific for CPPs or is a part of a general response elicited to external stimuli remains to be understood. An interesting scenario which is still speculative at this stage is that CPPs are recognized by the cell as generic exogenous agents since GO terms “pathogenic *Escherichia coli*” and “Epstein-Barr virus infection” appear to be among the most overrepresented terms in our KEGG Pathway analysis of upregulated genes after CPP treatment. However, full comparison of transcriptional profiles of cells treated with different extracellular stimuli would be needed to answer this question.

Among the most affected gene groups after CPP treatment are genes related to ribosome biogenesis. These alterations in gene expression appear to promote significant increase in nucleolar numbers and size, compatible with changes in translation and global protein synthesis. The nucleolus is a key player and central hub in sensing and responding to cellular stress. Ribosome biogenesis is one of the most energy demanding processes in the cell and therefore response to cellular stress is most commonly achieved by downregulation of rRNA synthesis and ribosome biogenesis [35]. As cells treated with penetratin display increased nucleolar size and increased expression of ribosome structural and biogenesis genes, treatment with CPPs per se may not be causing cellular stress but it may rather activate specific pathways affecting ribosome biogenesis, suggesting a possible impact on cell growth and/or proliferation. Dysregulated cell growth and proliferation are common in cancerous lesions and increased nucleolar numbers and size represent a prognostic indicator of malignancy [36]. On the other hand, changes in ribosome biogenesis and consequently, growth and proliferation, have also been associated with aging. Downregulation of certain ribosomal components leads to prolongation of lifespan of multiple organisms including mice and human [37,38]. The size of the nucleolus is also one of the determining factors in pluripotency and differentiation. Stem cells have bigger nucleoli which decrease in size and number after differentiation [39]. Depletion of certain nucleolar factors involved in ribosome biogenesis induces differentiation of pluripotent stem cells and their overexpression promotes reprogramming of somatic cells to pluripotency [40,41]. Therefore, manipulating expression of rRNA or ribosomal genes by CPPs may provide a paradigm to study alterations of cell growth and proliferation in several contexts such as cancer, ageing or differentiation.

Changes in growth and proliferation significantly affect cell cycle progression, including cell division. Increased expression of microtubule genes, decreased expression of centrosomal genes, reorganization of microtubules and decreased proliferation in cells treated with CPPs suggest that cells are going towards quiescence and differentiation which does not correlate with increased nucleolar size. However, although most differentiated cells have small nucleoli, neurons for example, have high levels of ribosome biogenesis serving as a supply of ribosomes for the local protein synthesis in developing neurites and numerous studies indicate that impaired ribosome biogenesis is a key feature of neurodegeneration [42,43]. We therefore suggest that, CPP treatment targets gene expression to induce ribosome biogenesis required for enhanced growth (but not proliferation) and that cells tend to enter a quiescent phase that exhibit lower proliferation levels, typical of differentiated cells. It would, therefore, be interesting to study CPPs on different cellular backgrounds and, for example, test the differentiation potential of stem cells or induced pluripotent stem cells upon CPP treatment. On the other hand, sustained CPP treatment over our time course may also lead to the induction of other regulatory pathways and different expression outcomes. Especially in the case of microtubules, different layers of regulation besides regulating the transcription take place. Most importantly, the microtubule dynamics is autoregulated by a negative feedback loop involving co-translational regulation of the stability of mature spliced tubulin pre-mRNA by unpolymerized tubulin [44,45,46]. For example, treating cells with microtubule-destabilizing drug cobretastatin A-4 (CoA-4) or with microtubule-stabilizing drug paclitaxel (PTX) does not change expression of tubulin genes but rather affects stability of mature tubulin mRNAs, diminishing in CoA-4-treated cells and increasing in PTX-treated cells [47]. Therefore, it is possible that a longer exposure to CPPs affects posttranscriptional regulation of microtubules rather than directly affecting their gene expression.

As part of the response to CPP treatment, we discovered a general decrease in expression of LncRNAs. This observation correlates with the suggested changes in growth and proliferation upon CPP treatment as majority of downregulated intergenic LncRNAs promote cell proliferation. One of the most downregulated LncRNAs in our dataset, MALAT1, has been widely reported to regulate the proliferation and migration of multiple cell types. In addition, MALAT1 overexpression increases the metastasis potential of cancer cells and is involved in the pathogenesis of various human diseases, involved in the onset and development of various tumors or tumor suppressor pathways [48]. In fact, knocking down MALAT1 reduces pancreatic tumor cell growth and proliferation both in vitro and in vivo [49]. At this stage we do not know how CPPs induce a reduction in LncRNAs expression, in particular MALAT1, and further studies are required. Understanding of these mechanisms can have important consequences as CPPs have been directly used for delivery of antisense oligonucleotides and siRNA against specific LncRNA to decrease their expression.

Taken altogether, our results suggest that treatment with cell penetrating peptides causes changes in expression of specific sets of genes related to ribosome biogenesis and microtubule dynamics which is accompanied with cellular phenotypes typical for quiescent cells. As DNA transcription is generally very fast and not so source-dependent as protein translation, altering expression of LncRNAs such as MALAT1 could be a way to rapidly control these processes. We speculate that CPPs have a direct role on cellular metabolism in particular key pathways related to growth and proliferation and open up the possibility of using them as direct tools in drug discovery design.

## Figures and Tables

**Figure 1 biomolecules-10-01567-f001:**
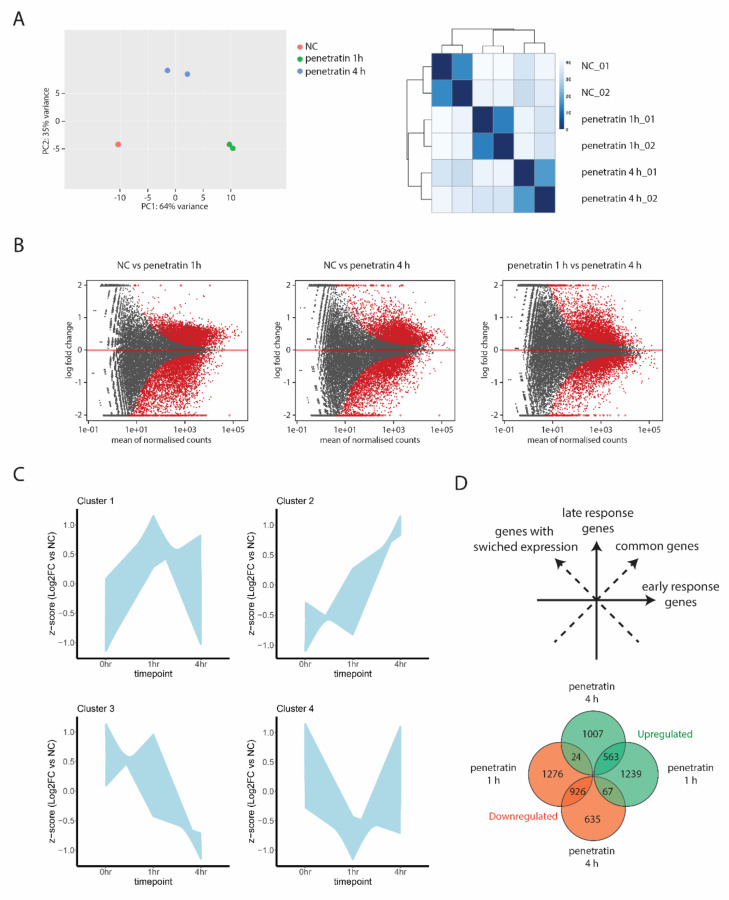
(**A**) PCA plot and distance heatmap of RNA-seq samples from WT Hela cells and cells treated with penetratin for 1 and 4 h. (**B**) The differences in gene expression between experimental conditions were visualized by MA plot by plotting the log2 versus the mean of normalized counts. Genes showed in grey do not show any significant change in their expression, while genes marked red are differentially expressed. (**C**) Plots of 4 different gene clusters showing up- or down-regulation after 1 or 4 h of penetratin treatment. y-axis shows z-score transformed log2 fold change relative to untreated negative control cells for each time point on the x-axis. (**D**) Distribution map of gene groups divided into early response genes (penetratin 1 h Up and Down), late response genes (penetratin 4 h Up and Down), common genes (intersection penetratin 1 h Up/penetratin 4 h Up, penetratin 1 h Down/penetratin 4 h Down) and genes whose expression is switched between different time points (intersection penetratin 1 h Up/penetratin 4 h Down, penetratin 1 h Down/penetratin 4 h Up. Venn diagram is showing number of genes found in each gene group.

**Figure 2 biomolecules-10-01567-f002:**
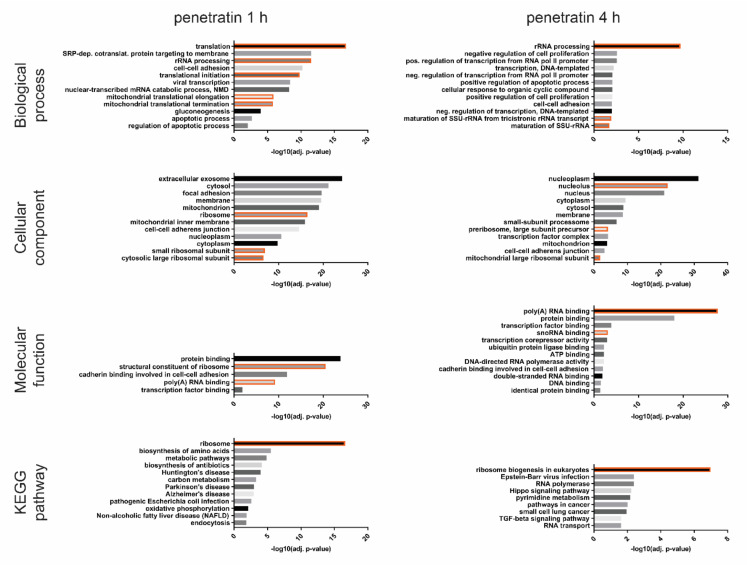
Gene ontology analysis of upregulated genes after 1- or 4-h treatment with penetratin. Gene ontology terms related to ribosome biogenesis and function are marked orange.

**Figure 3 biomolecules-10-01567-f003:**
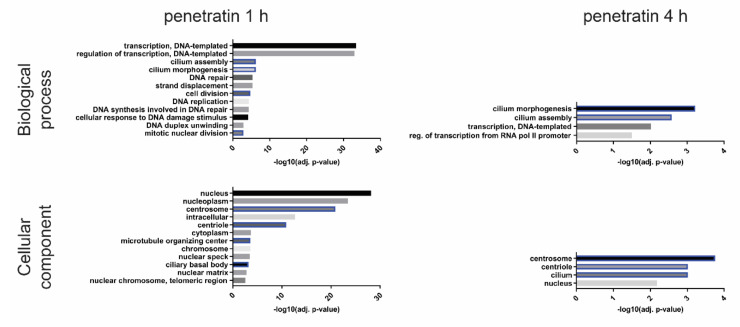
Gene ontology analysis of downregulated genes after 1- or 4-h treatment with penetratin. Gene ontology terms related to microtubules and centrosome structure are marked blue. Molecular function and KEGG Pathway GO annotation groups are not showed as they do not contain any relevant GO terms.

**Figure 4 biomolecules-10-01567-f004:**
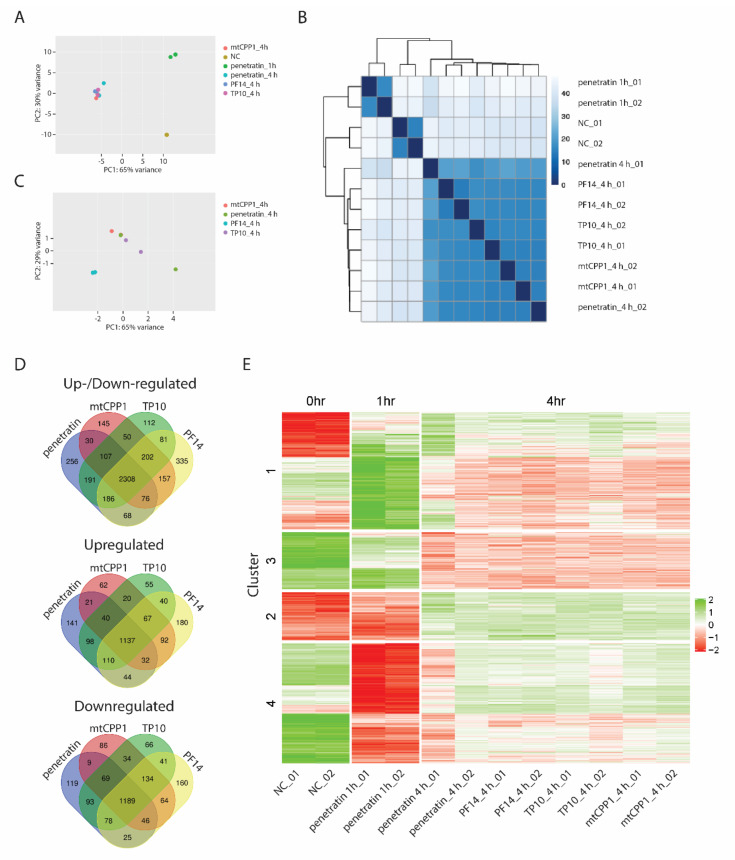
(**A**) PCA plot and (**B**) distance heatmap of RNA-seq samples from WT Hela untreated cells (NC), cells treated for 1 h with penetratin and cells treated for 4 h with penetratin, PF14, mtCPP1, or TP10, respectively. (**C**) PCA plot of RNA-seq samples from cells treated for 4 h with penetratin, PF14, mtCPP, or TP10. (**D**) Venn diagrams show intersection between differentially expressed genes in cells treated with different cell penetrating peptides for 4 h. (**E**) Heatmap showing gene expression changes induced after 1 or 4 h by different CPPs in gene clusters identified on the basis of Penetratin treatment. Scale bar shows z-score of normalized read counts (CPM).

**Figure 5 biomolecules-10-01567-f005:**
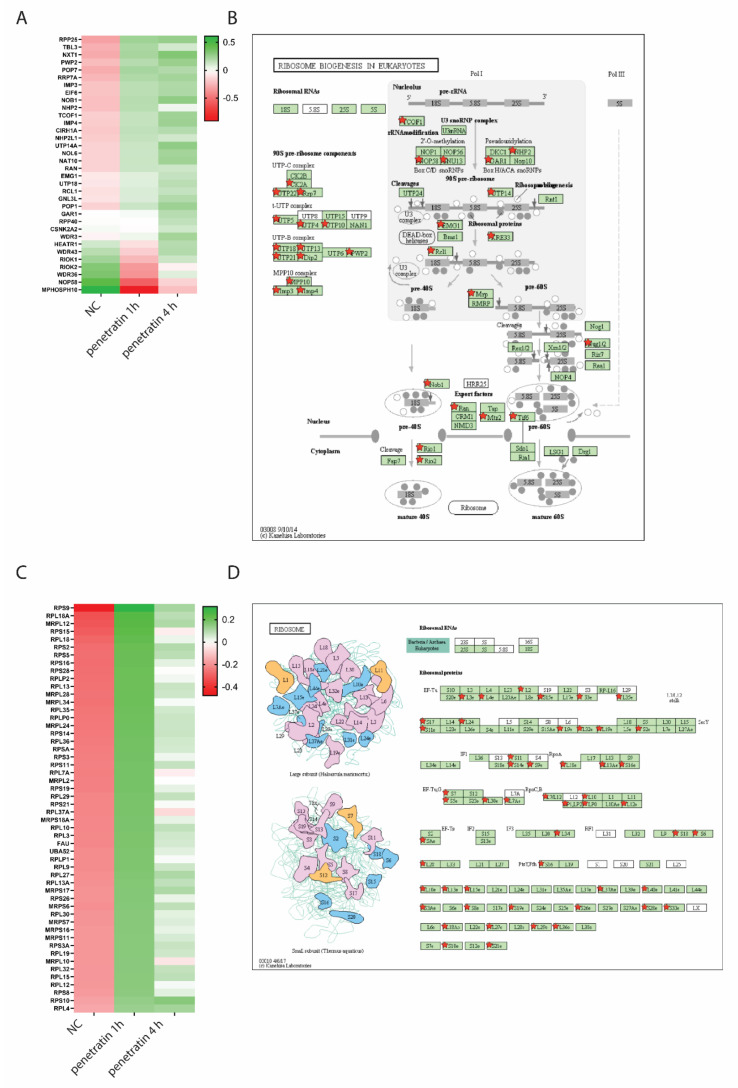
(**A**) Heatmap of differentially expressed genes related to GO-term “Ribosome biogenesis in Eukaryots”. Genes are organized in descending order based on the averaged log2(normalized read counts) for each sample. (**B**) KEGG-pathway map of genes related to “Ribosome biogenesis in Eukaryotes”. Genes marked with star are differentially expressed in penetratin treated samples. (**C**) Heatmap of differentially expressed genes related to GO-term “Ribosome”. Genes are organized in descending order based on the averaged log2(normalized read counts) for each sample. (**D**) KEGG-pathway map of genes related to “Ribosome”. Genes marked with star are differentially expressed in penetratin-treated samples.

**Figure 6 biomolecules-10-01567-f006:**
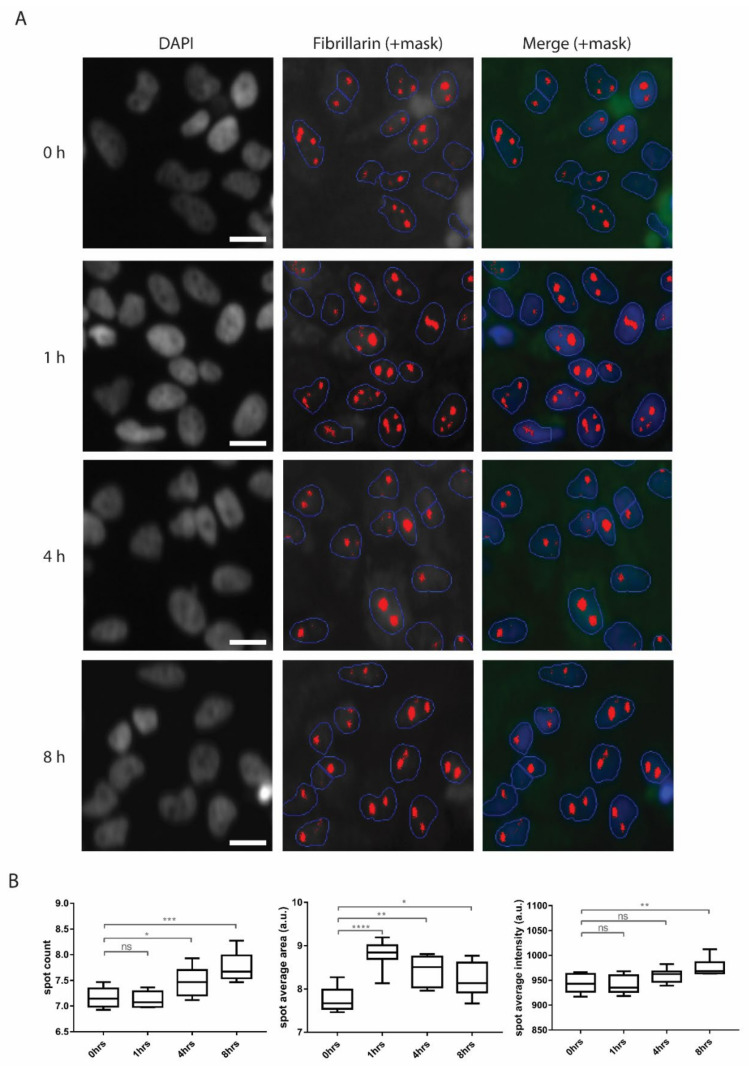
(**A**) Representative IF pictures of fibrillarin stained cells with a mask used for quantification by high-content screening (HCS) platform. Scale bar is 5 µm. (**B**) Quantification of HCS staining shows increase in size, intensity and number of fibrillarin-positive spots upon penetratin treatment. Significance: * *p* < 0.05, ** *p* < 0.01, *** *p* < 0.001, **** *p* < 0.0001, ns not significant.

**Figure 7 biomolecules-10-01567-f007:**
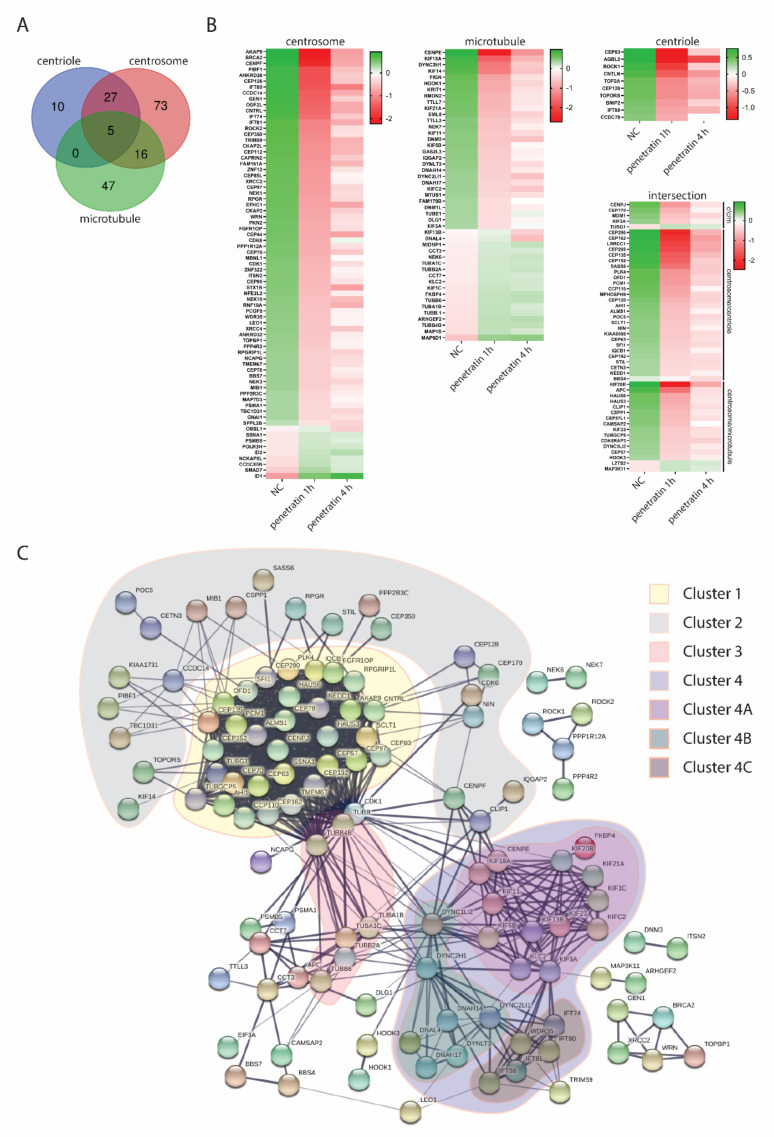
(**A**) Venn diagram of all differentially expressed genes related to gene ontology terms “Centriole”, “Centrosome” and “Microtubule”. (**B**) Heatmap of differentially expressed genes found in (**A**). Genes are organized in descending order based on the averaged log2(normalized read counts) for each sample. (**C**) Clustered interaction map of differentially expressed genes related to microtubules, centrosomes and centrioles.

**Figure 8 biomolecules-10-01567-f008:**
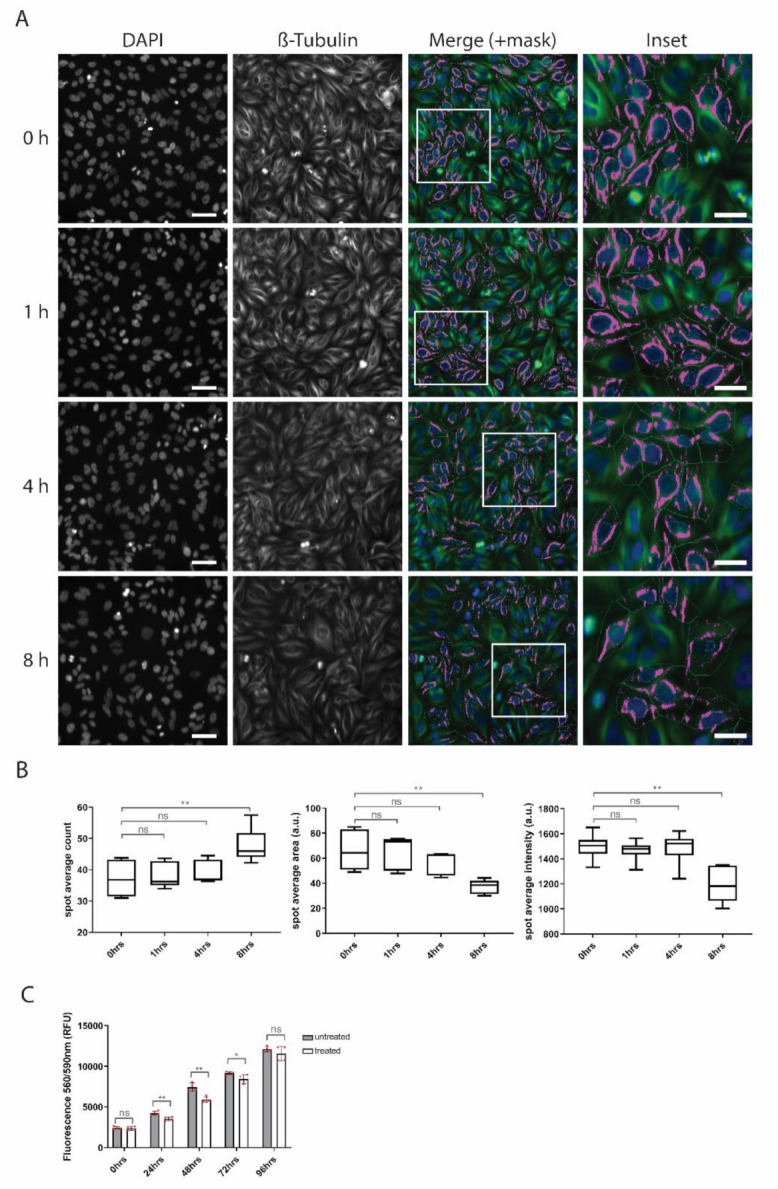
(**A**) Representative IF pictures of β-tubulin-stained cells with a mask used for quantification by high-content screening (HCS) platform. Scale bar is 20 µm. Scale bar in the inset is 5 µm. (**B**) Quantification of HCS of β-tubulin staining shows gradual decrease of spot intensity and area an increase in spot count. (**C**) CTB proliferation assay shows decreased proliferation rate upon penetratin treatment. Significance: * *p* < 0.05, ** *p* < 0.01, ns not significant.

**Figure 9 biomolecules-10-01567-f009:**
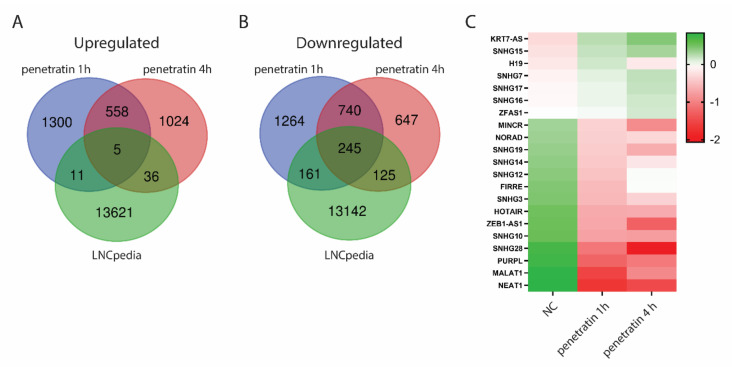
(**A**,**B**) Venn diagrams showing intersection between up- or down-regulated genes upon 1- or 4-h treatment with penetratin and the LNCpedia database of long non-coding RNAs. (**C**) Heatmap of selected LncRNAs found in the analysis. Genes are organized in descending order based on the averaged log2(normalized read counts) for each sample.

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
