# Peer review of "Transcriptional Profiling Reveals Ribosome Biogenesis, Microtubule Dynamics and Expression of Specific lncRNAs to be Part of a Common Response to Cell-Penetrating Peptides"

_biomolecules, 2020, doi:10.3390/biom10111567_

Round 1
Reviewer 1 Report
The authors addressed reviewers comments.
Reviewer 2 Report
The manuscript has been revised well. I have no further comments on the contents of the manuscript other than that some minor spelling errors are present.
I want to congratulate the authors on a well written paper and wish them the best for the future.
This manuscript is a resubmission of an earlier submission. The following is a list of the peer review reports and author responses from that submission.
Round 1
Reviewer 1 Report
Please see attached file

Reviewer 2 Report
The manuscript 'Transcriptional profiling reveals ribosome biogenesis, microtubule dynamics and expression of specific lncRNAs to be part of a common response to cell penetrating peptides' discusses in great detail the transcriptional changes occurring at different time-points after CPP treatment of HeLa cells. The paper is written in very concise manner and is overall well structured. Figures are well presented and informative. The data presented are relevant to the field and in line with the scope of the journal.
The manuscript should be accepted for publication with minor adjustments.
Minor comments:
- Figures should be checked for readability in printed format, some writing is very small (regards Figures 2, 3, 5, 6b-d, 7b, S1, S2)
- Introduction, results and discussion sections could be divided slightly better. E.g. p10 lines 295 to 302 and p13 lines 340 to 349 could be moved to the discussion. P15/16 lines 363 to 370 may be better in the introduction. These as examples, applies to other sections as well, generally where there are many references listed within the results section. Some of these results part are also repeated in the discussion.
- P16 lines 375 to 377: "The majority of LncRNAs belongs to the group of intergenic LncRNAs, followed by antisense LncRNAs, intronic LncRNAs and bidirectional LncRNAs". I would expect that this does simply reflect the amount of known lncRNAs in each of these groups, so the general genomic distribution of lncRNAs? This should be made clear in the text.
- a guilt-by-association analysis for the unknown lncRNAs found at different time-points could yield further information on whether these lncRNAs may also be involved in the described processes of ribosome biogenesis and microtubule dynamics. Such an analysis could be a nice addition to better understand the biology of lncRNAs that have not been studied in great detail.
- Using the DAVID classification tool, please state in the M&M section whether a standard background or a custom background was used. This can significantly affect outcomes. A custom background of genes expressed at baseline in HeLa cells should be preferred.